



# CO₂ emissions from German drinking water reservoirs estimated from routine monitoring data

**H. Saidi[1,2], and M. Koschorreck[1]**

 [1]{ UFZ - Helmholtz Centre for Environmental Research, Department of Lake Research, Brückstr. 3a, D-39114 Magdeburg, Germany}

[2]{ National Research Council – Institute of Ecosystem Study, Largo Tonolli 50, 28922 Verbania Pallanza, Italy }

Correspondence to: M. Koschorreck (matthias.koschorreck@ufz.de)

**Abstract**

Globally, reservoirs are a significant source of atmospheric $CO_2$. However, precise quantification of greenhouse gas emissions from drinking water reservoirs on the regional or national scale is still challenging. We calculated $CO_2$ fluxes for 39 German drinking water reservoirs during a period of 22 years (1991-2013) using routine monitoring data in order to quantify total emission of $CO_2$ from drinking water reservoirs in Germany.

All reservoirs were small net $CO_2$ sources with a median flux of 167 g C m$^{-2}$ y$^{-1}$, which makes gaseous emissions a relevant process for the reservoirs carbon budgets. In total, German drinking reservoirs emit 44000 t of $CO_2$ annually, which makes them a negligible $CO_2$ source in Germany. Fluxes varied seasonally with median fluxes of 30, 11, and 46 mmol m$^{-2}$ d$^{-1}$ in spring, summer, and autumn respectively. Differences between reservoirs appeared to be primarily caused by the concentration of $CO_2$ in the surface water rather than by the physical gas transfer coefficient. Consideration of short term fluctuations of the gas transfer coefficient due to variable wind had only a minor effect on the annual budgets. High $CO_2$ emission only occurred in reservoirs with pH < 7 and total alkalinity < 0.2 mEq l$^{-1}$. Annual $CO_2$ emission



correlated exponentially with pH, making pH a suitable proxy for $CO_2$ emission from German
drinking water reservoirs.

## 1 Introduction

Reservoirs are a globally important source of the greenhouse gases (GHG) $CO_2$ and $CH_4$ (St
Louis et al., 2000). Actually it is assumed that hydropower reservoirs globally emit 48 Tg C
as $CO_2$ and 3 Tg C as $CH_4$ (Barros et al., 2011). Existing studies on GHG emissions from
reservoirs focus on hydroelectric dams in boreal regions and the tropics and on dammed
rivers. Drinking water reservoirs in the temperate zone typically have a low trophic state and
GHG emissions are dominated by $CO_2$. Recent results indicate that they are a small source of
$CO_2$ to the atmosphere and can rather be a $CO_2$ sink during summer (Knoll et al., 2013).
However, existing $CO_2$ emission studies focus on few intensively studied reservoirs (Diem et
al., 2012; Soumis et al., 2004; Tremblay et al., 2005). Global inventories probably give a
realistic range of $CO_2$ emissions from surface waters (Raymond et al., 2013), but precise
quantification of GHG from drinking water reservoirs on the regional or national scale is still
challenging (McDonald et al., 2013; Seekell et al., 2014).
Upscaling is usually done by applying the thin boundary layer (TBL) approach (MacIntyre et
al., 1995). $CO_2$ exchange across the water surface is driven by diffusion and thus, regulated
by the concentration gradient between water and atmosphere and the physical gas transfer
coefficient K. K depends on the turbulence of the surface water. Although there is more and
more evidence that K is also influenced by convection (Read et al., 2012), in most studies it is
still derived from measured wind speed, using empirical equations (Cole and Caraco, 1998;
Crusius and Wanninkhof, 2003).
The concentration of $CO_2$ in surface waters is usually not directly measured but calculated
from two other measured parameters of the carbonate system, namely total inorganic carbon
(TIC), pH, or total alkalinity (TA). Thus, minimum data requirements are two parameters of
the carbonate system, water temperature, wind speed, and surface area. To obtain annual
budgets of $CO_2$ emission, both differences among reservoirs and temporal changes within a
system have to be considered. In practice, there is a trade-off between high monitoring
frequency and spatial coverage of numerous reservoirs because not all systems can be
monitored with high temporal resolution. Usually $CO_2$ concentration data are only available
for a few days of the year. Calculation of annual budgets from such sporadic measurements



may introduce systematic errors because high wind situations probably contribute
significantly to annual emissions (Morales-Pineda et al., 2014). By combining routine
monitoring data of $CO_2$ concentration from numerous German reservoirs with high temporal
resolution wind speed data from public weather stations we check, whether the low temporal
resolution of routine monitoring introduces a systematic bias in annual gas flux calculations.
The central aim of this study was to estimate the annual emission of $CO_2$ from Germany
drinking water reservoirs using data from routine water quality monitoring from a wide range
of reservoirs. By applying simple regression analysis we aimed to find out whether the $CO_2$
flux is primarily regulated by the gas transfer coefficient or by the $CO_2$ concentration. In
boreal surface waters, which are typically characterised by low alkalinity and high dissolved
organic carbon (DOC) concentration, the $CO_2$ concentration usually correlates well with the
DOC concentration (Jonsson et al., 2003; Whitfield et al., 2011), showing that aquatic
metabolism is a major driver of $CO_2$ oversaturation. In other regions, $CO_2$ in lakes seems to
be driven by DIC input from the catchment (McDonald et al., 2013). In high alkalinity lakes
in calcareous regions, $CO_2$ oversaturation is primarily caused by carbonate weathering (López
et al., 2011; Marcé et al., 2015). We used our dataset to get some information about the
principle drivers of the $CO_2$ flux from low DOC, low alkalinity waters, which are typical for
temperate drinking water reservoirs.

## 2   Material and Methods

### 2.1   Data source

We used a database containing routine water quality monitoring data from 39 German
drinking water reservoirs. Data were supplied by the reservoir operators and compiled in a
database in the framework of a research project about dissolved organic carbon in German
drinking water reservoirs (TALKO project). Available data span a period of 22 years (1991-
2013). Typical datasets for single reservoirs contained 10-20 years, the minimum period for a
single reservoir was 6 years with about monthly data. The data include both reservoirs and
pre-dams, which are characterised by a constant water level. A first quality control of these
data was performed using R statistic software. Typos, sign errors and rounding errors were
fixed using R functions. The dataset was checked for obviously wrong data by defining
minimum and maximum possible values.
Hourly wind speed data were provided by the German Meteorological Service (Deutscher
Wetterdienst) using the nearest weather station to each drinking water reservoir (Table S1).



The median distance between reservoir and corresponding weather station was 15 km (1 km –
38 km).

## 2.2 Calculations

The TBL approach (MacIntyre et al., 1995; UNESCO/IHA, 2010) was adopted to estimate
$CO_2$ fluxes from the reservoirs surface. This method uses semi-empirical equations to
calculate emission from concentrations of $CO_2$ in the surface water and the $CO_2$ exchange
coefficient. The flux J [mmol $CO_2$ $m^{-2}$ $d^{-1}$] of gas from water to air (diffusive emissions) was
calculated as the product of the gas exchange coefficient and the difference between gas
concentrations in surface water and air (Equation 1):
$J = K \times [CO_{2(water)} - CO_{2(air)}]$ (1)
Where
- $CO_{2(water)}$ is the concentration of $CO_2$ in surface water of the reservoir [µmol $l^{-1}$]
- $CO_{2(air)}$ is the concentration in air equilibrated water (calculated from the $CO_2$ partial

14       pressure in the air using Henry's law).

- K [m $d^{-1}$] is the gas transfer velocity approximated from the wind speed and

16       normalised to a Schmidt number of 600 (Crusius and Wanninkhof, 2003).

All calculations were done assuming a water density of 1 kg $l^{-1}$.

### 2.2.1 Surface water concentration of $CO_2$

Because of the best data availability, we calculated $CO_2$ from pH and TA, using the "seacarb"
package of R (Lavigne et al., 2014). Input parameters were water temperature, salinity =0,
depth=0, TA [mmol $l^{-1}$], and pH. For comparison, data were also calculated with CO2SYS
(Lewis and Wallace, 1998). Both tools gave the same results.

### 2.2.2 Concentration in air equilibrated water

We calculated the partial pressure of the gas in the water if it were in equilibrium with the
atmosphere ($CO_{2(air)}$ [mmol $l^{-1}$]) from the $CO_2$ partial pressure (p$CO_2$) in the ambient air
samples using Henry's law:
$CO_{2(air)} = P \div K_H$ (2)
For p$CO_2$ in the atmosphere (P [µatm]) we used hourly data of the atmospheric mixing ratio
of $CO_2$ [ppm] from the public monitoring station at Schauinsland (WMO World Data: Center



for Greenhouse Gases http://ds.data.jma.go.jp/gmd/wdcgg/wdcgg.html). This station is
located in the southern part of the Black Forest mountain range close to the top of mount
Schauinsland. It presents a reference site for the atmospheric background concentration in
Germany. The mixing ratios were converted to partial pressure by considering the altitude of
the particular reservoir:
$$P = mr \times P_{nn} \times e^{\frac{-alt}{scale\ height}} \times 10^{-6}$$  (3)
with mr being the $CO_2$ mixing ration [ppm], $P_{nn}$ = standard barometric pressure at sea level =
1 atm, alt = altitude of reservoir [m], and the scale height being 8500m. $K_H$ [atm l mol$^{-1}$] is
Henry's solubility coefficient for the actual water temperature.

### 2.2.3 Gas transfer velocities

There are several empirical expressions to derive the gas exchange coefficient (K) as a
function of wind speed and water temperature. We adopted the widely applied power function
presented in (Crusius and Wanninkhof, 2003):
$$K = \left[0.168 + \left(0.228 \times U_{10}^{2.2}\right)\right] \times \left(\frac{SC_{CO2}}{600}\right)^{-\frac{2}{3}or-\frac{1}{2}}$$  (4)
where
- $U_{10}$ is the wind speed at 10m height [m s$^{-1}$],
- SC is the Schmidt number for $CO_2$ (Wanninkhof, 1992):
$$SC_{CO2} = 1911.1 - 118.11 \times t + 3.4527 \times t^2 - 0.04132 \times t^3$$  (5)
Where t is the water surface temperature [°C].

### 2.2.4 Calculation of seasonal budgets

The temporal resolution of our data was heterogeneous. While gas transfer velocities could be
calculated with hourly resolution, $CO_2$ concentration data were typically available for 12 days
per year (4 to 293). To merge the data, we adopted 2 approaches:
a) "monthly" $CO_2$ fluxes were calculated by temporal upscaling of our measured data.
For each $CO_2$ concentration data point we determined the mean wind speed for the
same day and computed a daily mean flux for the day of sampling. For each month we
computed the mean of all available flux data within that particular month. If there





were no $CO_2$ data available for a particular month, we rejected that month from our
analysis.
b) For "hourly" $CO_2$ fluxes we assigned a $CO_2$ concentration for each wind speed data
point. We used the measured aquatic $CO_2$ concentration with the smallest time
difference to the particular wind data point.
Seasonal mean fluxes were calculated as: first, the means for each month were computed;
then the available monthly means were averaged within the following representative months:
spring (March-Mai), summer (June-August), and autumn (September-November). For annual
budgets the annual daily median flux was multiplied by 274 days, assuming that the $CO_2$
emissions are negligible during winter when reservoirs are ice covered. Summarised data for
each reservoir are provided in Table S2.

## 2.3    Statistical Methods

The statistical relationships between $CO_2$ evasion and different variables were calculated as
Spearman's linear correlations. Data were tested for log-normality by the Kolmogorov-
Smirnov test. To test for the significance of seasonal fluctuations we computed Tukey Honest
significant differences in conjunction with ANOVA. All statistical analyses were done using
R (R-Development-Core-Team, 2008).

## 3    Results

Surface $CO_2$ concentrations were between 0.002 and 11991 µmol l$^{-1}$. The annual median
concentrations in single reservoirs were mostly below 100 µmol l$^{-1}$, with a few reservoirs
having very high concentrations up to 2.4 mmol l$^{-1}$ (Figure 1a). The reservoirs were mostly
oversaturated with respect to $CO_2$. Under-saturation was observed between May and October
in 25 reservoirs. The median $CO_2$ concentration of all reservoirs was 72 µmol l$^{-1}$ (Table 1).
The reservoirs were typically exposed to low wind speeds, resulting in K values around 0.5 m
d$^{-1}$ (Figure 1b). The reservoirs could be grouped into low wind reservoirs, having a K below 1
m d$^{-1}$, and high wind reservoirs with k around 2 m d$^{-1}$.
If we consider all the seasons, we observed significant seasonal differences in $CO_2$
concentration (ANOVA test: $F_{2,1426}$=6.06, p= 0.002), fluxes (ANOVA test: $F_{2,234}$=3.72,
p=0.02) and gas transfer coefficient (ANOVA test: $F_{2,1426}$=8.48, p=0.0002). $CO_2$
concentrations were significantly higher in spring than in summer (Figure 2a, Figure S1a).
The gas transfer coefficient (resp. wind speed) was significantly higher in fall compared to the
other seasons (median 0.71 compared to 0.63 m d$^{-1}$) (Figure 2b, Figure S1b). Consequently,



fluxes were significantly lower in summer than in spring (Figure 2c, Figure S1c). Median
fluxes were 30, 11, and 46 mmol m$^{-2}$ d$^{-1}$ in spring, summer, and autumn respectively. Also the
variability of the flux was higher in spring and autumn.
We calculated annual $CO_2$ fluxes for each reservoir with and without inclusion of hourly wind
data. Both approaches gave similar results, but inclusion of high resolution wind data often
resulted in higher fluxes (Figure 3). For 27 out of 39 reservoirs the median annual $CO_2$ flux
was higher, for 7 reservoirs it was unchanged (less than 10% difference) while in 5 cases
fluxes calculated with hourly wind data were lower. The median $CO_2$ flux, however, was
hardly different between the two approaches (Table 1). An example dataset (Figure 4) shows
the effect of short periods of high wind speed on the flux. In this case, the annual median flux
was 71 and 132 g C m$^{-2}$ y$^{-1}$ without and with consideration of hourly wind speed data. The
median under-estimation for all studied reservoirs when not using high resolution wind data
was 22%.
On an annual scale, all reservoirs were a $CO_2$ source to the atmosphere (Figure 1c). By
multiplying the annual mean flux with the surface area we get the total annual flux from each
reservoir. The combined annual $CO_2$ flux from all reservoirs in our database was 13287 t y$^{-1}$
with a combined surface are of 35.56 km$^2$. If we assume a total surface area of all German
drinking water reservoirs of 118 km$^2$ (Köngeter et al., 2013), we can extrapolate a total $CO_2$
emission from all German drinking water reservoirs of 44091 t y$^{-1}$.
A simple regression analysis shows that the annual flux was regulated by the $CO_2$
concentration in the surface water rather than by the physical gas transfer (Figure 5). If we
analyse each reservoir separately, however, we observed significant correlations of the flux
both with $CO_2$ concentration and K. In 37 cases the flux was significantly correlated with $CO_2$
and in 32 cases with K. The fact that there were correlations between K and flux for single
reservoirs but not when all data are analysed together shows that the relation between K and
flux was reservoir specific.
Since the flux was correlated with the $CO_2$ concentration and the $CO_2$ concentration was
calculated from pH and Alkalinity, the $CO_2$ flux showed an exponential dependency on pH
(Figure 6a). High $CO_2$ fluxes only occurred in reservoirs with a median pH <6.5, which is the
dissociation constant of $H_2CO_3$ (Stumm and Morgan, 1981). The pH dependency can be
expressed by the following equation:





$J = 3.8573 + 5769.11406 \times e^{-\frac{pH - 4.94948}{0.63378}}$         (6)
We also observed a correlation with alkalinity with high median fluxes only occurring in
reservoirs with alkalinity below 0.2 µEq l$^{-1}$ (Figure 6b). On the other hand, there was no
relation between DOC and $CO_2$ flux (Figure 6c). There was a significant trend to smaller (by
area) reservoirs having higher $CO_2$ concentrations (Spearman rank correlation rho=-0.43, p=
0.006). For $CO_2$ fluxes there seemed to exist a similar relation, but the trend was statistically
not significant (rho=-0.23, p=0.1664).
**4    Discussion**
**4.1    $CO_2$ emission from German drinking water reservoirs**
German drinking water reservoirs are net emitters of $CO_2$ to the atmosphere. Our median $CO_2$
flux of 167 g C m$^{-2}$ y$^{-1}$ is high compared to the mean flux from hydroelectric reservoirs in the
temperate zone in the reviews of (St Louis et al., 2000) (150 g m$^{-2}$ y$^{-1}$) and (Barros et al.,
2011) (120 g m$^{-2}$ y$^{-1}$). A possible explanation is the high impact of stream water quality on the
drinking water reservoirs, caused by a typically low water residence time in the reservoirs.
Streams are known to be oversaturated with $CO_2$ (Raymond et al., 2013), with small streams
typically having higher $pCO_2$ (Hotchkiss et al., 2015). Because of better water quality,
drinking water reservoirs are preferably located in upstream areas with higher stream $pCO_2$.
This is supported by our observation of higher $CO_2$ concentrations occurring often in small
reservoirs, confirming earlier results (Raymond et al., 2013). It has been shown that the
gaseous $CO_2$ loss is linked to hydrology and shorter residence time increases surface carbon
loss (Striegl and Michmerhuizen, 1998).
Compared to typical $CO_2$ emission rates from temperate soils (745 ± 421 g C m$^{-2}$ y$^{-1}$, (Bond-
Lamberty and Thomson, 2010)) or a typical German forest site (-550 ± 91 g C m$^{-2}$ y$^{-1}$
(Grünwald and Bernhofer, 2007)), however, the area specific fluxes from drinking water
reservoirs are low. Considering further the small area of all German drinking water reservoirs
(0.03 % of German surface area), $CO_2$ emission from drinking water reservoirs is a negligible
$CO_2$ source in the national $CO_2$ inventory.
To investigate the significance of gaseous $CO_2$ exchange for the reservoirs carbon budget, we
estimated the total TIC content of reservoirs by multiplying the median TIC concentration
with the water volume of the particular reservoir for those eight reservoirs for which TIC data
were available. Total TIC inventories of reservoirs were between 1 t and 66 t resulting in





theoretical $CO_2$ residence times of 2 to 302 days. Thus, the annual $CO_2$ flux was of the same
order of magnitude as the TIC content of the particular reservoirs, showing that the gaseous
$CO_2$ flux was a significant process in the reservoirs carbon budget.
The observed seasonal pattern with low fluxes during summer is consistent with earlier
observations (Halbedel and Koschorreck, 2013; Knoll et al., 2013) and can be explained by
the seasonal stratification and depletion of $CO_2$ in the surface water due to primary
production, and increased surface concentration during autumnal mixing (Wendt-Potthoff et
al., 2014). Taken together, spring and fall contributed 87% to the annual $CO_2$ emissions. If the
focus is on the annual budget, we recommend to increase measuring efforts during the high
flux periods in spring and fall, on the cost of less intensive monitoring during summer.
Another information gap is winter. In winter, German drinking water reservoirs are usually
frozen, but the exact duration and timing of ice coverage is highly variable. $CO_2$ emissions
from non frozen reservoirs during winter would further contribute to annual emissions. To
improve the accuracy of annual budgets, the exact duration of ice cover have to be known for
each reservoir and year. Accumulation of $CO_2$ under ice is probably of minor relevance,
because water residence time in the reservoirs is low during high flow conditions in winter
and especially during snowmelt. Furthermore, our data give no hint on high $CO_2$
concentrations during early spring.
Our median K of 0.70 m d$^{-1}$ is virtually identical to the global average for lakes and reservoirs
estimated from global wind data (0.74 m d$^{-1}$ (Raymond et al., 2013)). It is well known that the
determination of K from wind speed is prone to some error, especially at low wind speed
(Crusius and Wanninkhof, 2003). The location of the weather station represents another
source of error. All the weather stations used for the reservoirs with high k-values are located
in more wind exposed crests. Four of the "high K reservoirs" were caused by the weather
station Zinnwald-Georgenfeld which is located at 877 m a.s.l. in the Ore Mountains. Since the
reservoirs are located in valleys, $CO_2$ fluxes in the "high K reservoirs" are probably over
estimated. A way to circumvent this problem would be the determination of reservoir specific
correction factors for the wind speed. Considering the uncertainty related to the
representativeness of the wind data from public weather stations for the reservoirs, the use of
a constant K might introduce only a minor error. Applying a constant K of 0.7 m d$^{-1}$ results in
a median $CO_2$ emission from all reservoirs of 107 g m$^{-2}$ y$^{-1}$, which is 28% lower than the
median flux calculated using monthly wind data. We interpret this as an estimate of the error
caused by the non-representative location of weather stations. However, considering the





observed low dependency of the flux on K, uncertainty in the determination of K is probably
not a serious problem for our upscaling approach.

## 4.2    Effect of short term wind fluctuations

We found a significant under-estimation of the total annual $CO_2$ flux by 22% when only
considering wind data from the day of which we also had $CO_2$ concentration data. This was
because we missed some high wind periods, especially in fall, which contributed significantly
to the annual flux. Even if the local wind at the reservoir was not perfectly represented by the
weather stations, this conclusion is justified, since the probability for storm events was
probably comparable at the reservoir and corresponding weather station.
Our mean error of 22% is most probably a conservative estimate because recently it has been
shown that wind does not only directly influence K but due to enhanced surface mixing also
affects the surface concentration of $CO_2$ (Morales-Pineda et al., 2014). Storm events can also
affect $pCO_2$ by flushing $CO_2$ from the catchment into the lake (Vachon and del Giorgio,
2014). In our case the error was highly case specific. Some reservoirs even showed an
opposite effect, most probably because low wind periods were more frequent. Thus, an
analysis of typical wind patterns at a particular reservoir should allow to predict whether the
inclusion of high frequency wind data have the potential to significantly improve the $CO_2$ flux
estimate for a particular site.
Besides periodic changes in wind speeds and storms, there exists a typical diurnal wind
pattern at the reservoirs in our study. Wind is increasing during the day and then calms down
around sunset and during the night. This diurnal pattern is included in our simple approach,
since we used the daily mean wind speed for the low resolution flux calculation. The use of
wind data obtained during water sampling by hand-held wind meters, a common practice in
many studies, most probably overestimates the daily $CO_2$ flux, because low wind periods
during the night are not considered. However, wind is not the only factor causing diurnal
pattern. Recent research indicates that night-time cooling causes convective mixing near the
surface and thus, may enhance gas fluxes during the night (Eugster et al., 2003; Read et al.,
2012). Neglecting this effect is probably the main reason for the commonly poor
parametrisation of K at low wind-speed (Cole and Caraco, 1998) and would result in an
under-estimation of the real flux. Our study does not consider the effect of convection on K
and thus, our annual budgets are probably conservative estimates. The role of convection and
a better parametrisation of K for upscaling deserve further research.





### 4.3    Regulation of the $CO_2$ flux
### 4.3.1  $pCO_2$ versus K
The difference in the $CO_2$ flux between reservoirs was primarily caused by the concentration
of $CO_2$ in the surface water rather than by the physical exchange coefficient K. This was
caused by the higher between reservoir variability of $pCO_2$ compared to K. Thus, to quantify
the annual flux in an unknown reservoir, high frequency monitoring of the $CO_2$ concentration
is more important than increasing the quality of the wind data. Since the surface $CO_2$
concentration in the reservoirs is probably predominantly determined by inflow water quality
rather than reservoir internal processes, $CO_2$ emissions are probably largely regulated by
catchment processes (Stets et al., 2009). This confirms studies showing that the $CO_2$ emission
from lakes may be controlled by catchment productivity (Maberly et al., 2013) or carbonate
weathering (Marcé et al., 2015). Catchment processes and inflow water quality are obviously
more important than hydrodynamics in regulating the annual $CO_2$ emission from German
drinking water reservoirs. The major effect of reservoir internal processes seems to be the
reduction of the $CO_2$ flux during summer, caused by stratification and primary production
(Halbedel and Koschorreck, 2013). However, the effect of this flux reduction in summer is at
least partly compensated by enhanced fluxes in fall because then $CO_2$ from the hypolimnion is
mixed to the surface. Because of the highly dynamic nature of these mixing processes, high
frequency monitoring of $CO_2$ would increase the precision of the flux quantification
especially in fall.
Besides these seasonal fluctuations, the $CO_2$ concentration can also fluctuate diurnally, driven
by photosynthesis during the day. Thus, the daytime of sampling should have an influence on
the quality of the $CO_2$ data. We consider this effect less relevant in our case, since routine
water samples are taken during normal working hours, when $CO_2$ concentrations are probably
intermediate.
### 4.3.2  No correlation with DOC
Our results confirm earlier studies that the aquatic $pCO_2$ in temperate lakes and reservoirs
does not depend on the DOC concentration (McDonald et al., 2013). This is in contrast to
observations in boreal lakes and tropical waters (Borges et al., 2015), where often a
correlation between DOC and $pCO_2$ has been observed. One reason could be that our DOC
concentrations (Table 1) are low compared to a global average of lakes ($7.6 \pm 0.2$ mg $l^{-1}$).
Boreal lakes typically contain even higher DOC concentrations (Sobek et al., 2007). More





probable, however, is that any effect of DOC is masked by the influence of TIC import from
the catchment and pH effects (López et al., 2011). There is no simple link between lake
metabolism and annual $CO_2$ flux. The net annual $CO_2$ flux cannot be used to judge whether a
reservoir is net heterotrophic or not, since the flux is both influenced by TIC transport and
metabolism (Stets et al., 2009).
**4.4    pH as a proxy for the $CO_2$ flux**
The $CO_2$ concentration in the reservoirs clearly correlated with pH, confirming results from
Knoll et al., who found that the pH was the best predictor of $pCO_2$ in the Midwestern
reservoirs they studied (Knoll et al., 2013). A similar correlation between diffusive $CO_2$ flux
and pH has been observed in 151 Danish lakes (Trolle et al., 2012), 948 Florida lakes
(Lazzarino et al., 2009), and several reservoirs (Alshboul and Lorke, 2015; Halbedel and
Koschorreck, 2013; Quiñones-Rivera et al., 2015; Soumis et al., 2004). Thus, pH dependency
of the $CO_2$ flux seems to be a general observation in temperate surface waters. These results
also highlight the importance of precise pH measurements for accurate surface water GHG
budgets (Herczeg and Hesslein, 1984). Because of its logarithmic nature, pH is especially
prone to analytical error. This is critical when using routine monitoring data for $CO_2$
calculations.
The pH is a result of alkalinity (mainly influenced by catchment geochemistry) and TIC
(influenced both by catchment processes and aquatic metabolism) (Marcé et al., 2015; Müller
et al., 2015). Especially between pH 5 and 7 even small changes in $CO_2$ significantly alter the
pH. This effect is less relevant in DOC rich boreal lakes, which are often acidic, and in
eutrophic lakes, were primary production shifts the pH to high values. In acidic waters the
situation is complicated by the fact that organic acids contribute to alkalinity (Abril et al.,
2015). This could be relevant in our high-emission reservoirs, which are all low in alkalinity
(Figure 6b). Regardless the underlying mechanisms, the strong correlation with pH suggests
the use of pH as a proxy for the $CO_2$ flux for modeling or upscaling. We calculated the $CO_2$
flux from each reservoir from its mean pH ($J_{pH}$) using equation 6 (Fig. S2). The resulting
median $CO_2$ flux of was virtually identical to the flux obtained from our monthly data (Table
1). As a rule of thumb, relevant $CO_2$ emissions do only occur in reservoirs with pH<7 and
alkalinity below 0.2 µEq $l^{-1}$.
The surprisingly good fit to pH can be partly explained by the rather uniform and low
alkalinity values. Larger differences in alkalinity would result in more variable $pCO_2$ at
similar pH values. Thus, the use of pH as a proxy for $CO_2$ emissions might be applicable only



under low DOC, low alkalinity conditions as they are typical for German drinking water
reservoirs.

## Acknowledgements

This work was financially supported by the BMBF (project TALKO, BMBF 02WT1290A)
and by the European Community (COST action ES1201 (NETLAKE)). We thank Michael
Opitz for providing the Talko database and Rafael Marcé for constructive comments on an
earlier version of the manuscript.

## Author contribution

H. Saidi analysed the data and wrote the manuscript. M. Koschorreck designed the study and
wrote the manuscript.



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



**Tables**
**Table 1.** Descriptive statistics of annual median data from all reservoirs K=gas transfer
coefficient, $J_{hourly}$=$CO_2$ flux calculated using high resolution wind data, $J_{monthly}$=$CO_2$ flux
calculated using mean wind, $J_{pH}$=$CO_2$ flux calculated from mean pH of each reservoir (n=39,
for DOC and pH annual means).

|  | min | max | median | mean | SD |
|---|---|---|---|---|---|
| $CO_2$ [$\mu$mol l$^{-1}$] | 15 | 2365 | 72 | 283 | 523 |
| K [m d$^{-1}$] | 0.5 | 2.17 | 0.7 | 0.9 | 0.5 |
| $J_{hourly}$ [g C m$^{-2}$ y$^{-1}$] | 14 | 7386 | 167 | 765 | 1545 |
| $J_{monthly}$ [g C m$^{-2}$ y$^{-1}$] | -3 | 6710 | 148 | 689 | 1518 |
| $J_{pH}$ [g C m$^{-2}$ y$^{-1}$] | 20 | 6271 | 146 | 769 | 1442 |
| DOC [mg l$^{-1}$] (n=19) | 0.92 | 6.15 | 3 | 3.18 | 1.44 |
| pH | 4.9 | 8.7 | 7.3 | 7.05 | 0.98 |





# 1  **Figures**

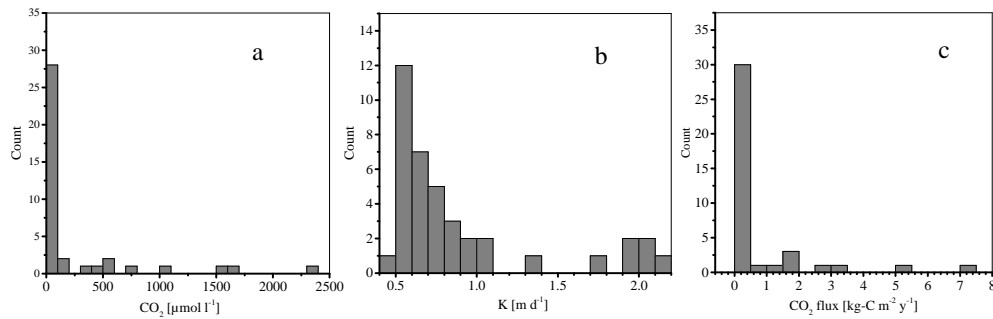

3  Figure 1: Histograms of median annual data of the different reservoirs: a) $CO_2$ concentration,

4  b) K, c) $CO_2$ flux. A Kolmogorov-Smirnov test showed that the data were not log-normal

5  distributed.





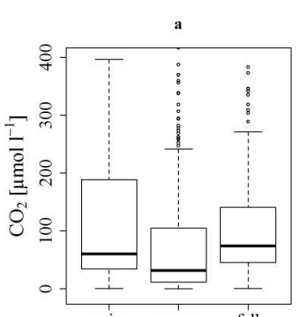 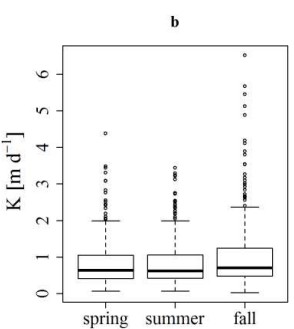 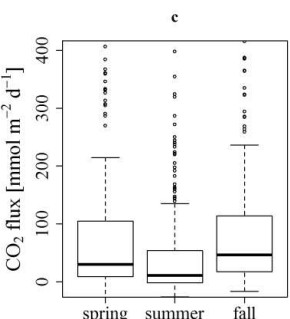

2  Figure 2: Boxplots describing the seasonal fluctuation of $CO_2$ concentration (a), K (b), and

3  $CO_2$ flux (c). Data points are the mean seasonal data for each reservoir (n=39). Extreme

4  values (higher than 400 $\mu mol \, l^{-1}$ and 400 $mmol \, m^{-2} d^{-1}$) are outside the plots.





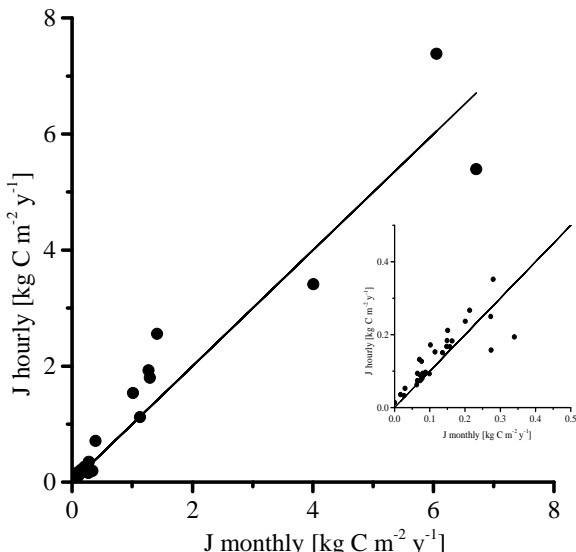

2    Figure 3: Median annual $CO_2$ flux for different reservoirs calculated on an hourly or monthly

3    basis. The insert shows a magnification of the left part of the graph.





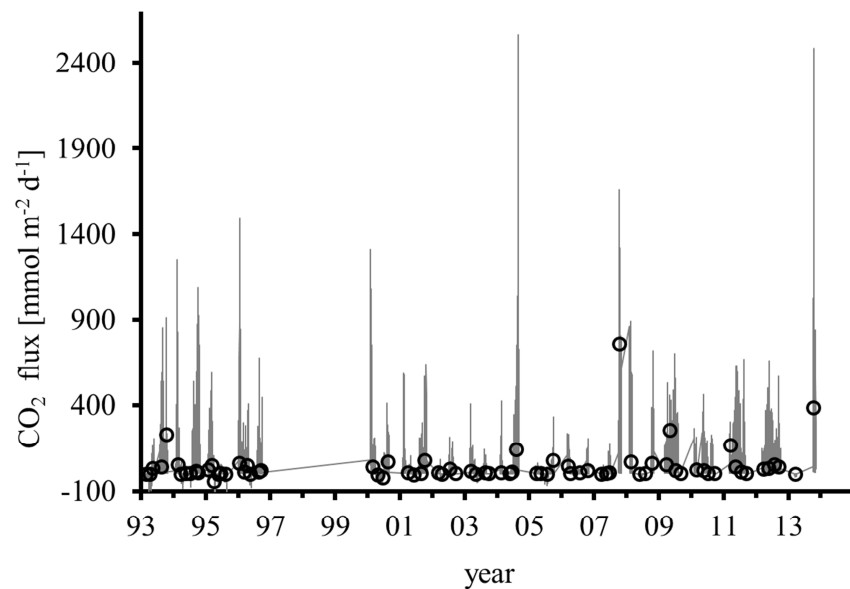

2 Figure 4: CO$_2$ flux from Rappbode pre dam calculated with (line) and without (circles) using

3 high frequency wind data.




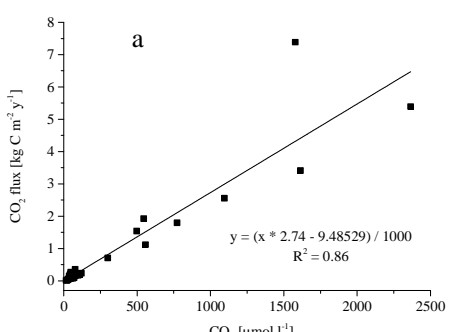
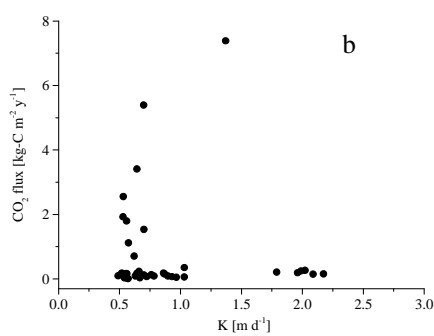

2 Figure 5: Dependence of the annual median $CO_2$ flux on the annual median $CO_2$

3 concentration (a) and K (b).





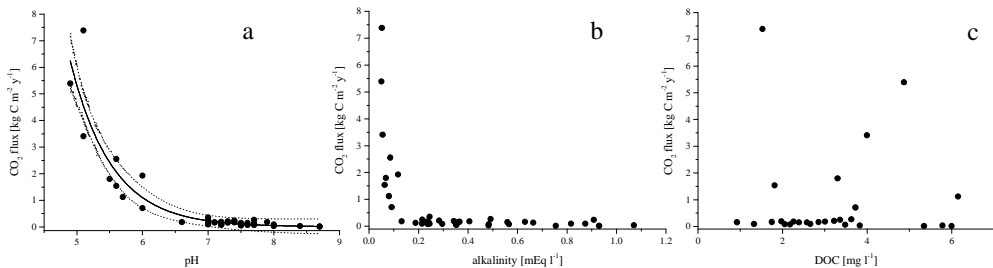

2   Figure 6: Dependence of the median $CO_2$ flux on a) mean pH, b) alkalinity, and c) mean DOC

3   concentration in the different reservoirs. Lines in a) show an exponential fit with 95%

4   confidence interval ($R^2 = 0.85$).