# Peer review of "CO2 emissions from German drinking water reservoirs estimated from routine monitoring data"

_Biogeosciences, 2015_

## Referee Comment (RC1) · Anonymous Referee #1 · 20 Jan 2016

Review of Saidi and Koschorreck CO2 emissions from German drinking water reservoirs estimated from routine monitoring data submitted to BG

In this paper, the authors use temperature, pH, and Total alkalinity (TA) monitoring data combined with wind speed data in order to calculate CO2 emissions from drinking water reservoirs in Germany. These reservoirs, represent all together a modest surface area of about 110 km2 for the whole Germany, that is, equivalent to that of Lake Müritz in Germany, but five times lower than lake Constance. One of the conclusions is that CO2 emissions from these drinking water reservoirs are negligible. The authors also attempt to relate the calculated CO2 fluxes with measured variables, and conclude that pH can be a good proxy for CO2 emissions.
[Figure]

I have two different types of criticism for this MS: one concerns the objectives of the study that is not well defined, and the speculative character of the interpretation of biogeochemical processes of these specific systems. The other one concerns the use of correlations between parameters that are not independent, but that in fact were calculated one from the other (as for instance the $CO_2$ flux is calculated from pH) and some imprecisions on important aspects of aquatic chemistry.

First, the emissions being very small, this research would interest a large audience only if the MS could provide a detailed and original description of the question of drinking water in a broader biogeochemical context. The MS lacks from a clear definition of the objective of the study (other than using an available dataset, which at the moment appears as the only motivation for this work): is the topic here "drinking water $CO_2$ emissions", is it "all reservoirs $CO_2$ emissions"? "All inland water emissions?" When reading the MS it was not clear why focussing on drinking water reservoirs: if they are not significant $CO_2$ source, at least are they special reservoirs? It also took me some time to understand that in Germany all these reservoirs receive only surface water and that the water is stored in reservoirs before being treated to make drinking water. With such high DOC values I guess this is not drinking water, and drinking water cannot contain live phytoplankton either as assumed in the MS. What's the interest in temperate region in storing surface water before treatment ? Most importantly, is it a general procedure found elsewhere? How do $CO_2$ emission vary in reservoir as a function of the nature of the surface water used? What happens to water when pumped between the stream and the reservoir? How does this compare with other countries that use groundwater for drinking? And finally, how will water treatment affect the $CO_2$ emissions? In the introduction and the Discussion, comparison is made indifferently with other reservoirs (hydroelectric or irrigation), or with natural lakes, without considering the specificity of each type of systems in terms of carbon source, or other biogeochemical drivers. This makes the problematic confusing. No time course of measured pH and TA and calculated $pCO_2$ are shown, no relationship is found with water residence time, stream water characteristics etc. . . This makes the MS poor in terms of scientific

interest.

Second, the authors perform correlation between non-independent parameters that where calculated from each other. They calculate first the pCO2 in the water (using pH and TA); and, second, FCO2, the CO2 flux at the water-air interface (using the water-air pCO2 gradient and the wind-speed parameterization of Crucius and Wanninkhof 2003). No direct measurement of FCO2 (that would have allowed the computation of K and thus an independent comparison with wind speed) were performed in this study. Later in the MS, correlations are found for instance between FCO2 and pH, leading to the conclusion by the authors that pH can be defined as a "proxy" of FCO2. This is a truism. Indeed, given the broad range in pH values, such observation is almost trivial, because low pH generates high calculated pCO2 (that might in part be overestimated due to organic alkalinity), and thus cannot be used to predict anything, and does not constitute a scientific advance. Indeed, this correlation might be driven in parts by the presence of organic acids that generate an overestimation of the calculated pCO2 at low pH. The maximum value of calculated CO2 concentration was 11990 micromol L-1 (Page 6 L 19), that is a pCO2 more than 300 000 ppmv. Can such high values be affected by the bias described by Abril et al. (2015)? How much the occurrence of the bias in pCO2 calculation does affect the correlation between calculated pCO2 and pH ? and the global estimation of the CO2 flux as well.

There are also several uncertainties and imprecision in the way the gas transfer velocity was calculated, sometimes using wind speed data measured in the mountain to calculated gas exchange in reservoirs located in the valley more than 800 meters below (P9 L24-27) and no fetch effect (size of reservoirs) is considered in the parameterization as a function of wind speed.

Figures 3 and 5 have little interest as they correlate parameters that are not fully independent.

Detailed comments In the intro P2L8-12 Why do drinking water reservoirs have a low

trophic state? First paragraph mixes drinking water and other reservoirs. What are their differences?

P2L24. In fact direct measurements of CO2 concentrations exist. The problem of pCO2 calculation in low pH low alkalinity waters should be mentioned in the intro. P3L3. The concentration of CO2 was not monitored. Monitored parameters were pH and TA.

P3L8 "by applying simple regression analysis. . . or by the CO2 concentration." You don't need to make such "simple regression" because FCO2 is explicitly a function of CO2 concentration and wind speed. L13 "in other regions. . . catchment". DIC input from catchment is important whatever the region. P8L13 "possible explanation is the high impact of stream quality on the drinking water reservoirs, caused by typically low water residence time". This statement is vague. This should have been discussed with much more details. CO2 emissions should have been analysed in terms of surface water origin and residence time of water in reservoir. L16 : "because of better quality, drinking water reservoirs are preferably located in upstream areas with higher stream pCO2. This is supported by our observation of higher CO2 concentrations occurring often in small reservoirs". This is very speculative: are small reservoirs necessarily connected to small streams and large reservoirs to large rivers ? I would make an alternative speculation: CO2 concentration is higher in small reservoir because they have lower fetch, which limits gas exchange. Do the authors have evidence for their interpretation being less speculative than mine?

---

## Referee Comment (RC2) · Anonymous Referee #2 · 15 Feb 2016

This study investigates CO2 emission from German drinking water reservoirs, based on calculations of pCO2, estimations of the gas transfer coefficient, and the surface area of all German drinking water reservoirs.

While the study seems quite straightforward as to the data and methods, I have several severe concerns with this study. In its present state, I do not think it constitutes a valuable addition to Biogeosciences.

First, none of the aspects, findings or conclusions in this study are new. The only added knowledge is calculations of CO2 emission from a particular kind of reservoirs from one region, and fluxes were comparable to other reservoirs and other regions. Maybe because of that, the introduction does not manage to convincingly outline the

purpose of the study, and a hypothesis is lacking. This very much limits, in my opinion, the usefulness of this study.

Second, several of the conclusions and interpretations are not supported by the data. For example, the discussion of the importance of hourly CO2 emission dynamics is only based on hourly wind data (from met stations quite distant from the reservoir, i.e. not at the reservoir), but no hourly resolution in pCO2 data (assumption of constant pCO2 to calculate hourly flux). Another example is the interpretation that input of CO2 with river inflow is a more important CO2 source that internal production, which can not be made unless a C budget is established or internal production measured.

Thirdly, wind speed, which was used to estimate k from the Crusius & Wanninkhof 2003 empirical relationship, was measured at stations that were quite distant from the reservoirs (and maybe even on the top of mountains, as mentioned in the discussion). Hence, the resulting k may not be very representative of the k at the reservoir surface, making the resulting CO2 emission fluxes questionable. This applies particularly to the calculation of hourly CO2 emission; if the wind speed data are not representative for the site of the reservoir, the calculation of hourly fluxes is meaningless.

Lastly, the study concludes that since pH strongly correlated with CO2 emission flux, it can be used as a proxy for upscaling. Since the authors calculated pCO2 from pH and alkalinity, the correlation is simply a result of chemical equilibrium, and the correlation is based on Y variable that depends very strongly the X variable. This is statistically questionable. In addition, CO2 emission is estimated from pCO2 (in this study calculated from alkalinity, pH and temperature) and k; in this study, alkalinity and k were rather constant, leaving a strong influence of pH on CO2 emission. Any significant changes in either alkalinity or k would therefore very much weaken the relationship between pH and CO2 emission. This makes its use for upscaling very limited. Also, as the authors state, pH has very strong leverage on calculated pCO2 within a certain pH interval, and any uncertainties in pH measurement result in corresponding uncertainty in pCO2, further questioning the use of pH as a proxy for upscaling CO2 emission. I

therefore think that the statement to use pH as a proxy for upscaling CO2 emission is not substantiated, and may even be misleading researchers during future studies.

In addition, there are a few other aspects that in my opinion are problematic.

P1L20. This sentence invokes the impression that carbon budgets were established, which was not the case.

P2L17. The thin boundary layer approach is used to calculate CO2 exchange with the atmosphere, not for upscaling.

P2L24. While I agree that the majority of inland water pCO2 data in the literature are calculated and not directly measured, I think using the term "usually" is not appropriate. Direct measurement is vastly preferable to calculation of pCO2, given the strong effect of pH on the results, and particularly at low alkalinity

P5L24. It seems what was done here was interpolation rather than upscaling.

P6L10. Can these reservoirs in Germany really be assumed to be ice-covered during whole winter? Many lakes in Germany never freeze over. Maybe this argument could be backed up with water temperature data?

P6L19. The range in pCO2 is very wide, spanning 7 orders of magnitude. This seems, from my experience, unrealistic, and points towards that there may be outliers that could be attributable to the pCO2 calculations. Was there a cutoff for low alkalinities applied?

P8L11. The calculated median CO2 emission is comparable to the cited values, given the uncertainty in calcualtions and assumptions, not high.

P9L15. In many reservoirs, sedimentation is high, and degradation of sediment organic matter might be an important source of CO2 in reservoirs, maybe particularly during low flow (i.e. low DIC input from catchment) in winter. Low CO2 in spring does not indicate absence of under-ice accumulation since outgassing at ice-out is very rapid

and may not be captured by the sampling program.

P9L30. These calculations illustrated how important the gas exchange velocity k is when calculating CO2 emission flux. So indeed, flux is highly dependent on k, contrary to what is stated in this paragraph. It is evident from this paragraph of the discussion that with the available input data, the calculation of hourly CO2 emission is not warranted. Therefore, the entire section 4.2. is not relevant.

P11L9 and L 13. These statements are not supported by data.

P12L15-26. This statement is highly questionable, given the pH has such a strong leverage on calculated pCO2 (a very small change in pH can give a big change in pCO2), and that pH, in fact, is difficult to measure with high accuracy, particularly in soft waters. In addition, the observed relationship between pH and CO2 emission of this study is caused by chemical equilibria and near-constant k and alkalinity; in situations where k and alkalinity vary more strongly, the relationship would become much weaker. Giving a recommendation to use pH for upscaling could be highly misleading for future studies.
* * *

---

## Referee Comment (RC3) · Anonymous Referee #3 · 16 Feb 2016

This manuscript reports a study using drinking water reservoir data from 39 reservoirs in Germany to estimate CO2 emissions. This is, to the best of my knowledge, one of the first papers to report on any greenhouse gas emission from drinking water reservoirs; therefore, the data is valuable to have in the literature. However, the paper requires major rewriting of sections that includes (1) a more thorough introduction, (2) better explanations in the methods and results, (3) deletion of the pH results/discussion and (4) the inclusion of a conclusion. With significant major revisions, the manuscript could be considered for publication; however, there is nothing new to be gained from this paper other than CO2 emissions from drinking water reservoirs. Perhaps the authors can come find a storyline to tell that would make this dataset more interesting for

publication.

General comments: Language: Recognizing that English is likely not your mother tongue, I commend you on the grammar throughout the paper. However, I urge you to take advantage of this time to expand your English scientific writing skills. A good example to do so is by trying to make slightly longer, more eloquent sentences instead of very short ones. I have pointed out some examples in specific comments below. First paragraph of introduction can be expanded upon. Discuss the importance of GHG emissions from inland waters in general and their contribution to carbon cycle. Then reservoirs and how they are man-made impoundments and thus would be considered anthropogenic sources of GHG. Next about the focus on hydropower reservoir GHG emissions because they are supposed to be a green source of energy. Lastly, any information about GHG emissions from drinking reservoirs, which I am guessing there are not a lot, and why it's important to also measure that. These subjects could easily be 3 paragraphs of the introduction. Then go into what is usually measured in drinking water reservoirs and how it can help estimate CO2 emissions from them but the data can be sporadic (basically P2, L25 starts this paragraph). Then a paragraph about potential drivers of CO2 emissions (the things you will test for, like DOC; P3, L7-13). Then end with a paragraph describing the aim of your study. Methods: The methods need a lot more work to be comprehensive. The structure is also odd. More data is need for Section 2.2.1. Section 2.2.2 needs to be corrected a bit and given more description. Section 2.2.3 needs more description. Section 2.2.4 is highly confusing and needs much more description and better structure. See specific comments for more details. The results need a lot of restructuring and explanations as well. The most confusing part is understanding what fluxes, calculated how, were being used for the results. I also do not agree with the pH and alkalinity relationships with CO2 flux as being meaningful as those two parameters were used to calculate CO2 in the first place. The only way to meaningfully make any statements regarding the use of pH as a proxy for CO2 is if you have independent measurements of both CO2 and pH. Therefore, the section in the discussion regarding this is also an issue. The paper needs a conclusion

paragraph of some type. It ends too abruptly as is. I suggest a discussion of how drinking water reservoirs could and should implement CO2 emission monitoring into their normal routines.

Specific comments: P1, L19 – I assume you mean that this median flux implies it's a relevant process for the carbon budget of each individual reservoir. Is that right? P1, L19-21 – Move the sentence 'In total, German drinking reservoir emit 44000 t of CO2 annually...' to the last sentence of the abstract P2, L3 – Mention different types of reservoirs here in the first sentence – the ones that St. Louis mentioned. P2, L5-7 – Move the sentence 'Existing studies on GHG emissions...' to the second sentence of the paragraph and you must have an example reference for each type of reservoir you mention (hydro in boreal, hydro in tropics, dammed rivers) P2, L4 – preface this sentence by saying that 'Hydropower reservoirs have a been a central focus of GHG emission studies from reservoirs as any emissions of these gases would counter the 'greenness' of this type of energy supply.' P2, L7-8 – You need a reference for this sentence about drinking reservoirs. If the reference is Knoll et al. 2013 then you should move that citation to the first sentence P2, L10 – I do not understand the point of this sentence: 'However, existing CO2 emission studies focus on few intensively studied reservoirs'.. please clarify P2, L13-14 – why is it still challenging? P2, L15-25 – I don't think this information regarding methods is necessary for your introduction. You are not really discussing these particularly methods as a bias for data calculation and interpretation. You are mostly concerned about P2, L25-32 – Start the next paragraph with a discussion about what is usually measured in drinking water reservoirs and how that can be used to estimate CO2 emissions, but that the resolution is heterogeneous so annual budgets are difficult to come up with. P2, L32-P3, L2 – this sentence belongs with the last paragraph of the introduction that describes what you aim to do in this study P3, L5 – don't say 'By applying simple regression analyses'... more like 'We aim to find relationships that help explain...' P3, L7-L13 – all of this information about potential drivers of CO2 emissions deserves it's own paragraph above P3, L11 – DIC is not defined yet P3, L18 – Mention some of the 'routine water quality monitoring

data' particularly the ones you used for your calculations. I am guessing some of them are in Table S2, so you should also reference that Table here, but clearly make it now Table S1 as it is appearing before the table you reference on Line 29. P3, L24-27 – These few sentences starting with 'A first quality control. . .' are not necessary. P3, L29 – Change to Table S2 if you are adding the other table reference above as suggested. P4, L3 – change to 'fluxes from the reservoir surfaces.' P4, L6-7 – replace 'the difference between gas and concentrations in surface water and air' with 'the CO2 concentration gradient' P4, L10-14 – don't list these as bullets. Place them in normal lines separated by ';' P4, L12 – list the units P4 L15 – add a sentence after the last sentence about density that states: 'All calculations and procedures to determine the variables for the basic flux equation are described next' P4, L17-20 – Give more details about these equations/R packages and how CO2 is actually calculated. There were no measurements of actual dissolved CO2 or pCO2 in the datasets? P4, L20 – If you used the seacarb package definitively then add to the end of the last sentence 'Both tools gave the same results, but we decided to use seacarb because. . ..' Why did you decide on that one? P4, L25 - change 'P' to 'pCO2' . . . and type the equation properly (although I imagine the editor will do this): ãĂŰCOãĂŮ_2(air) = ãĂŰpCOãĂŮ_2/K_H P4, L26 – move sentence P5, L5-6 where you define KH to just after you state equation 2; Then discuss how you derive pCO2 P4, L26 – Just call the variable pCO2 P4, L26 and P5, L4 – define the 'mixing ratio' somehow P4, L30 – 'It represents a reference site' . . .. Change 'presents' to 'represents' P5, L3 – do you have a reference for this equation? P5, L13-14 – place these variable definitions in the text after 'where' on L12 P5, L13 – were all of the met stations taking wind speed at 10m? P5, L14 – give a quick reason why the Schmidt number is necessary and that it is based on temperature. You do not need to define the equation P5, L15 – delete this equation – not necessary P5, L11 – You need to add at the end of this section an explanation of the exponents (-2/3 or -1/2) and how and why you use chose which one to use. I am guessing you used something similar to Wanninkhof 1992 which was based on Jahne 1987 where -1/2 is used for wind speeds under 3 m/s and -2/3 used for wind speeds over 3 m/s. P5,
[Figure]

L17 – Calculation of seasonal budgets: I find this section a bit confusing. You are not only calculating monthly, hourly, seasonal fluxes here, but also daily fluxes. And I don't understand dhow you 'merge data'. You should start with the highest resolution and move to lowest... so hourly, then daily, then monthly, then seasonal, then annual. And be explicit at each step and give them proper names and then use those names later on because it's highly confusing in the results to know what you used for what. P5, L19 – CO2 concentration here being the concentration in the water? P5, L19-20 – I do not understand how you have 'typically available for 12 days per year' but you have a possibility from 4 to 293 days? I would add 'days' in those parentheses. But what is the 12 day number? Is that an average? P5, L22-23 – You state that you 'determined the mean wind speed for the same day and computed a daily mean flux' – but how many hours of wind speed of the hourly data did you use to determine the wind speed? Be specific. I would also state that you 'we determined the mean wind speed from the hourly data using XX hrs' P6, L1-3 – So you had hourly wind data and then you tried to find the closest CO2 concentration data to each hour? But you had at best maybe daily data but apparently not that often if you have a range of 4 to 293 and an average of 12 days in the year. I do not think that an hourly flux is then reasonable to calculate. What were the smallest time differences? P6, L4 – rewrite: 'Seasonal mean fluxes were calculated by finding the means for each month, then the available monthly means...' P6, L7 – these daily fluxes are calculated from those used to make the monthly data, right? P6, L8 – Were these reservoirs always ice-covered? I don't believe they could be and their ice on periods likely varied quite a bit. Do you have any data on this? P6, L14 – replace 'done' with 'conducted' P6, L17 – use scientific notation for '11991' and you can use '$\mu$M' for the units. These number 0.002 and 11991 don't match Table 1. Should they? P6, L18 – replace 'single' with 'individual' and change units P6, L19 – use scientific notation with units of $\mu$M for 2.4 mmol/L P6, L19-21 – combine these two sentences 'The reservoirs were...' and 'Under-saturation was observed...' and give the saturation value of CO2 in $\mu$M in the sentence too. P6, L20 – so since you observed undersaturation then you should have had some uptake fluxes (i.e., negative

[Figure]

CO2 fluxes), yet I do not see those in Fig. 1c. Why not? P6, L21 – instead of saying '25 reservoirs', please give % of reservoirs. NOTE: Do this for every time you discuss how many reservoirs were like this or that. Use % and not number of reservoirs. P6, L21 – this median concentration is of all measurements – make that explicit P6, L22-24 – give wind speed values for the low and high winds. Stating that the K values were mostly 'around 0.5 m' is not very precise. Use some sort of statistic to state this. In fact, most values actually look above 0.5. I would give % here for things as well. Figure 1 – change y-axes to % instead of count. Log the CO2 concentration and the CO2 flux to better see the distribution. The fluxes shown in 1c are median annual fluxes, but how were these annual fluxes calculated? P6, L25 – delete 'If we consider all the seasons' and just start with 'We observed. . .' and be explicity that these were the seasonal budget calculated from monthly means. P6, L29 – what does 'resp. wind speed' mean? P6, L30 – what season does the 0.63 m/d refer to? Both spring and summer? If so, be explicit P6, L30-31 – Rewrite: 'Consequent of both low CO2 concentrations and low K, fluxes were lowest in summer.' P6,L31 – P7, L1-2 – Combine these two sentences 'Median fluxes were. . .' and 'Also the variability. . .' P7, L3 – using the terms 'with' and 'without inclusion of hourly wind' is not useful here, I think. Use the suggestions made earlier in the methods about how to label each resolution of calculation and stick with it throughout the paper. P7, L4-5 – rewrite and combine with sentence from P7,L11-12 about the underestimation: 'Both approaches gave similar results with hourly fluxes slightly higher and a consequent 22% underestimation using monthly-based data on average' and put a R2 here. Figure 3 – are those 1:1 lines in the figures? Put in the caption if they are P7, L5-7 – change the count of reservoirs to %. And rewrite: 'Hourly-based median annual CO2 fluxes were higher than monthly-based median annual CO2 fluxes in XX% of reservoirs, while in XX% of reservoirs the values were within 10% of each other and in XX% of reservoirs hourly-based were lower.' P7, L13-16 – Begin this paragraph with the second sentence describing how you calculated the total flux. Then combine the first and third sentences into one. I am confused how you calculated these values. You have multiple years of measurements from reservoirs so how do you

account for this in calculating the total per year, especially when you don't have a value for every year for every reservoir? Once you clear that up and make it explicit, then also explain how you calculated the mean annual flux – such as did it mean that you also had negative fluxes and you averaged those as well? What about instead of using an annual mean flux, you used your monthly fluxes, calculated a monthly emission rate in tons/month and then added those up to get the yearly loss. Would you get a different total emission rate? Which would be more representative? P7, L17-18 – how did you extrapolate to all of the other reservoirs you didn't measure? Please explicit. P7, L19-25 – This paragraph has a good point but it's not clearly written. Make it obvious in the first sentence that you used the mean annual flux from each reservoir (is this right? That if a reservoir had 10 years of measurements then you would take the average of that for this Figure 5?). In the second sentence, state what resolution calculation you are looking at for each individual reservoir. Do you have a figure for these? And use % instead of reservoir counts. Do you think the fact that the individual reservoirs have a correlation with K is dependent on the resolution of calculation you used – for example if you used the hourly than you have better wind data? Or even if you used the daily then you have a daily wind speed rather than an average for the all year. I cannot determine if the K correlation is real for individual reservoirs or an artefact of how you made your calculations. Figure 5 – try logging the x-axis P7, L26-31 – Since you did calculate CO2 with pH and Alkalinity data, it is obvious and expected that you would see a relationship between CO2 flux and those parameters. I believe this makes this correlation a bit circular, not valid, and not worthy of discussing. Equation 6 is thus also not very useful. P8, L2-6 – The discussion of a DOC relationship is valid however. P8, L10 – why did you choose this value instead of the monthly? Why not use a range of 148-167 and you are closer to St Louis value. Also cite Table 1 again here. P8, L11 – should look like this: 'in the reviews of St. Louis et al. (2000) and Barros et al. (2011) with values of 150 and 120 g m-2 y-1, respectively.' P8, L14 – rewrite 'Streams are known to be oversaturated with CO2 (XX) with small streams (i.e., lower order) typically. . .' P8, L15-16 – rewrite: 'Drinking water reservoirs are preferably located in

upstream areas close to headwaters to ensure high quality drinking water and are thus receiving water from lower order streams that typically have higher pCO2.' P8, L17-18 – rewrite: 'This is supported by our observation of the highest CO2 concentrations occurring in the smallest reservoirs.' P8, L18-20 – I don't understand what you are trying to say with this sentence. Please clarify P8, L21-26 – Area and areal flux doesn't matter as much in this case as does the relative totals of different sources of CO2 emissions. Is there a better reference to use here to discuss the implication of drinking water reservoir emissions? Also, you are citing a negative flux in a German forest site and say that the drinking water emissions are low. That doesn't make any sense. And you need a ref for the drinking water reservoir surface area of 0.03% P8, L28 – is TIC defined earlier? P8, L30-31 – You say 'total TIC inventories of reservoirs' – does mean only the 8 reservoir for which you have data? If not, then clarify how you extrapolated. The residence times of 2 – 302 days is based on what exactly? The flux of 167 g C/m2/d x the 1t to 66 t range over the area??? Explain better. P9, L1 – how do you know the annual CO2 flux was of the same order of magnitude as the TIC content? Please clarify P9, L6 – the increased surface concentration during autumnal mixing was because of the CO2-rich hypolimnion being mixed upwards in fall, correct? You should state this a bit clearer here. P9, L9 – change to 'in spring and fall at the cost of less. . .' P9, L13 – change to 'the exact duration of ice cover has to be.' P9, L14 – you state that the accumulation of CO2 under ice is probably unimportant but there are studies about this. State some. If it turns out it could be important and you just don't know, that is fine but you must state that. P9, L15 – why are there high flow conditions in winter under ice? P9, L22 – use the uppercase K P9, L23 – get rid of quotes around 'high K reservoirs' P10, L11-12 – rewrite: 'not only directly influence K but also CO2 as a result of enhanced surface mixing.' P10, L16-17 – rewrite: 'should all the prediction of whether the. . . wind data has the potential. . .' P10, L24 – delete comma before 'because' P10, L25 – add 'a' before 'diurnal' P10, L27 – rewrite 'and thus enhances gas.' P10, L28 – rewrite 'Neglecting convective mixing is.' P10, L31 – delete comma after 'thus' P11, L3-4 – rewrite: 'The variability

in the... caused by CO2 concentration in the surface.' P11, L4-5 – avoid using 'This was' to start a sentence. That type of beginning can be very ambiguous. Be specific. Consequently, I don't really understand your point with this sentence. Please clarify. P11, L6-7 – it is not necessarily more important to get CO2 concentration than wind data. I would rather say they are both equally important, especially if you have no idea of the wind conditions. Fluxes are in fact considerably enhanced in high winds. P11, L7-9 – instead of 'Since the' use 'We suggest that surface CO2' and continue 'We suggest that surface CO2 concentration... determined by inflow water quality (i.e., CO2 from inflowing streams) rather than internal processes, which implies that CO2 emissions are largely regulated by catchment processes.' P11, L10 – again, do not begin with 'This'... rewrite: 'Other studies had similar results showing that...' P11, L12-14 – Why are catchment processes and inflow water quality more important? Because there was no relationship seen with K? Be specific P11, L14-15 – rewrite: 'Internal processes seem to have the largest effect during summer when CO2 fluxes are lowest, likely due to primary production.' P11, L17- rewrite: '...because CO2 accumulated in the hypolimnion during stratification is...' P11, L22 – rewrite 'by light-dependent photosynthesis; thus sampling time has an influence on the...' P11, L28-30 – rewrite ' .. DOC concentration (REF), which is in contrast...' and delete the rest after the Borges reference. P11, L30 – rewrite: 'One explanation for this is that our DOC...' P12, L2-5 – There is not simple link – that is true. But you also don't have a lot of measurements either of the other processes so that is why you cannot truly judge the contribution of the reservoirs to the carbon balance. You should state that more explicitly P12, L7–12 – I have a real issue with this discussion as the correlation found in your own data is not valid. The only way to determine if pH is a proxy for CO2 is to independently measure both of those parameters and compare and not calculate one based on the other. P12, L18-30 – The first half of this paragraph is useful to support your use of pH to calculate CO2. But then you begin discussing your pH-CO2 flux relationship again, which you cannot do because again you calculated CO2 based on pH. P12, L31-P13, L2 – This could be combined with some of the above info to support your use of pH to calculate

[Figure]

CO2, but that is all. Figure 6 – I don't believe you can use the relationship of pH and alkalinity with flux as they are used in the calculations. Figure S1 – Cannot read the axes at all of this figure.

―――――――――――――――――――

---

## Author Comment (AC1) · 1 Mar 2016

Reply to reviewer 1:

We would like to thankfully acknowledge Referee 1 for providing valuable comments that will significantly contribute to the improvement of our paper. It was indeed our principal goal to get a first estimate of CO2 emissions from German drinking water reservoirs – not reservoirs in general or a worldwide perspective. This was mostly driven by the fact, that the literature is somewhat biased towards boreal lakes and boreal and tropical hydropower reservoirs. Doing this we faced some practical problems (mostly linked to data availability) like: how to do temporal interpolation? How to deal with heterogeneous data frequency? Is it a problem if no on-site wind data are avail-

able? We thought getting such data from drinking water reservoirs and addressing these practical issues would be interesting enough to be published as a paper. Our data do not really allow a discussion about the role of drivers, global upscaling or the general biogeochemistry or function of drinking water reservoirs or drinking water treatment. From the reviewers comment we conclude that our objectives are probably not interesting enough to attract an international audience. We could include some further analysis of possible drivers like trophic state or catchment characteristics into our paper. However, such further data analysis would need a complete new story and a completely new manuscript. It was our hope to have this first paper covering methodology of flux determination and providing a nation-wide upscaling. We would prefer to publish the detailed discussion of drivers like water quality or catchment characteristics in a future second paper.

The obvious problem that the resulting fluxes were calculated from pH values and are thus not independent from pH was raised by all reviewers. Actually it is clear that the pH is mainly controlled by $pCO_2$ in these waters. It is also clear that the flux is also calculated from and thus, not independent from wind speed. However, our results show that inter-reservoir variability was primarily governed by $pCO_2$ (and thus, pH) and not wind. This is because variability in $pCO_2$ was higher than variability in wind speed. We think that this is useful information and not meaningless, as stated also by the other reviewers. It shows that the exact quantification of k600 is probably less important when comparing different reservoirs. We consider this point also important, because, the correlation with pH was extremely good and because although a similar dependence was observed by several other studies, this point has rarely been addressed. Studies which correlate $CO_2$ fluxes with lake or catchment variables frequently come up with the observation, that pH was the best predictor of the $CO_2$ flux. We think it is worthwhile to discuss the usefulness of this finding with respect to upscaling. As correctly stated by reviewer 2, the exact shape of the flux-pH curve depends on alkalinity and k of the particular systems. Thus, our relationship of course cannot be directly transferred to other systems. In a revised manuscript we would do a more detailed comparison with

similar curves from the literature to explore the transferability of our results. A weak point is indeed that we did not proper address the possible role of organic acids. In a revised manuscript we would try to do some error estimates based on our DOC data, pH and literature information. We would like to keep the pH relation in the paper, but since it was criticized by all reviewers, we would put less emphasis on it.

Another important point, also raised by the other reviewers, is the quality of our wind data. Of course it would be nice to have site specific calibration of the wind speed–k600 relation, wind data measured on each reservoir and direct pCO2 measurements. We are trapped here somehow between detailed case studies which have all these data and global studies which sometimes even use a mean wind speed or wind data obtained from models. Since there are no on-site long term wind data available for all our reservoirs, we see no better method than the one we applied. We consider this not a serious problem, because we could show that considering short term wind fluctuation would not change our results dramatically. We may have a systematic over estimation of the flux due to an over estimation of wind speed due to the location of our weather stations. The uncertainties related to wind data are already addressed in the discussion section. In a revised version of the manuscript we would try to better quantify the possible error resulting from our wind data. However, an in depth discussion of this issue is probably not very productive given the generally observed poor parametrization of U10 at low wind and the non-consideration of thermal advection. The regulation of k600 needs to be studied in more detail in case studies, which then could address also points raised by the reviewer, like influence of fetch or reservoir size.

---

## Author Comment (AC2) · 1 Mar 2016

reply to referee 2

The reviewer raised similar points as reviewer one. Here we address those points which are not included in our reply to reviewer 1.

To discuss the issue of diurnal fluctuations of $CO_2$ we would need high frequency $pCO_2$ data. Such data are available for a number of well studies systems. We also have such data from one of the studied reservoirs. It does not make much sense to address this point when relying only on the monitoring data presented in this manuscript. In a revised version we would include an error estimate based on assumed reasonable

daily pCO2 amplitudes.

The reviewer is right that our discussion of the relative importance of inflow versus reservoir internal processes is not supported by data. In a revised version of the manuscript we could use water quality data from the reservoir inflows to assess the effect of catchment export on pCO2 in the reservoir.

The sediment is indeed a source of CO2. However, this translates not in a direct effect on the surface flux. Our reservoirs are stratified during most time of the year and sediment derived CO2 accumulates in the hypolimnion over summer. Part of this CO2 gases out after autumn mixing. This effect is accounted for in our study and is the reason for elevated fluxes in autumn.

The reviewer is probably right that we miss some emissions directly after ice off. In a revised manuscript we will analyze this point by having a closer look on winter pCO2 data.

---

## Author Comment (AC3) · 1 Mar 2016

Reply to reviewer 3

We thank reviewer 3 for this really careful review. We are also thankful for the language corrections. Germans are used to rather long sentences and we are always trained that writing short sentences is good stile in scientific English. Here we address those points which are not included in our replies to the other reviewers. If not stated different in this reply we will follow the reviewers advices. We will widen the introduction and elaborate better our hypothesis. We will also widen the method section, especially the part about budget calculations. Our different ways to calculate fluxes obviously caused some confusion. In a revised manuscript we would focus on monthly data,

because that scale fits best to the temporal resolution of our CO2 data. We would then use the hourly data only to estimate the potential error when ignoring short term wind fluctuation.

P1, L19: Yes, we will add "individual"

P2, L10: In this and the following sentences we would like to express that there are case studies about single reservoirs but not studies covering a larger number of drinking water reservoirs.

P2, L13-14: Upscaling on a regional/national scale is challenging, because it is difficult to find the right balance between detailed information on particular sites and using simplifications and means. In global studies, for example, you may apply mean wind speeds on a larger area. In detailed case studies you need wind measured continuously on the reservoir. What is the best way on a regional scale?

P5, L13: Yes – wind speed data were measured at 10 m above open flat ground.

P5, L19-20: The temporal resolution of our data was quite heterogeneous. For one reservoir we indeed had nearly daily data for some time. For one reservoir we had weekly data. The most common case was that we had about one data point per month. 12 is the average data point number per reservoir and year.

P6, L1-6: We think it is justified to calculate hourly data even if only few annual pCO2 values are available. From continuous measurements of pCO2 in a reservoir we know, that diurnal fluctuations of are not very high in these nutrient poor systems and short term "jumps" of pCO2 are rare. Thus, we think we are doing a minor error when using the pCO2 measurements as being representative for a period of some weeks. It was one of our main aims to check whether ignoring short term wind dynamics results in different results. That was also the reason why we first present the low temporal resolution data, then switch to high resolution (to check that question) and finally went for an annual budget. However, in a revised manuscript we would calculate our annual

budget from the monthly rather than hourly data.

P6, L8: No, reservoirs were not always ice covered. Unfortunately, ice data on ice coverage are not readily available. We will also add this point as a suggestion: It would be nice to have the information about freezing data or ice off date for all the reservoirs.

P6, L17: The $CO_2$ concentrations given here do not match table one, because here we state the maximum and minimum values of all our combined datapoints. In Table one, we compare annual median data for particular reservoirs.

P6, L20: Under saturation is not visible in fig.1c, because the figure shows annual median values. As stated somewhere else, all reservoirs were annually net $CO_2$ emitters.

P6, L30: The median for spring and summer was both 0.63 m/d

P7, L13-17: We will check the alternative approach to calculate annual fluxes proposed by the reviewer

P7, L19-25: We used the median annual flux as mentioned in figure caption (not the mean annual as mentioned by the reviewer). Yes - in Fig.5 we took the median of the available yearly data for each reservoir. As stated above, in a revised paper we will primarily rely on monthly data. Thus, we have to recalculate Fig.5 (which is actually based on hourly calculations). We think that the fact that the fluxes from individual reservoirs have a correlation with K is independent from the resolution (hourly or daily wind speed). Just four hourly resolution we have some higher fluxes. The K correlation, if it is verified for individual reservoir, is real and it is not depending on the way of calculation.

P8, L10: We think that the hourly calculated values are probably nearer to reality, because they consider wind fluctuations.

P8, L18-20: We want to acknowledge the work of Striegl and Michmerhuizen who found that shorter residence time (a feature also of small reservoirs) favors $CO_2$ emission.

P8, L30-31: We calculated a hypothetical residence time of CO2 by dividing the TIC content by the CO2 flux. We will explain that more specificly in the text.

P9, L15: We will check discharge data in our database to verify high flow in winter

P11, L4-5: We want to say that the flux was different between reservoirs primarily because the CO2 concentration was different.

P11, L6-7: We agree that both CO2 concentration and k are important to know. However, as our comparison of calculation with and without high resolution wind data shows, ignoring hourly wind fluctuations gave only 22% different total annual fluxes. Thus, uncertainties in k are probably less critical.